# Infantile-Onset Isolated Neurohypophyseal Langerhans Cell Histiocytosis with Central Diabetes Insipidus: A Case Report

**DOI:** 10.3390/children9050716

**Published:** 2022-05-13

**Authors:** Mizuki Tani, Shota Hiroshima, Hidetoshi Sato, Kentaro Sawano, Yohei Ogawa, Masaru Imamura, Makoto Oishi, Keisuke Nagasaki

**Affiliations:** 1Department of Pediatrics, Niigata Prefectural Shibata Hospital, Niigata 957-8588, Japan; apisto.7824@gmail.com (M.T.); totsutotsu0118@gmail.com (H.S.); 2Division of Pediatrics, Department of Homeostatic Regulation and Development, Niigata University Graduate School of Medical and Dental Sciences, Niigata 951-8510, Japan; sho980522@gmail.com (S.H.); sawano@med.niigata-u.ac.jp (K.S.); yohei_oga@yahoo.co.jp (Y.O.); mimamura@med.niigata-u.ac.jp (M.I.); 3Department of Neurosurgery, Brain Research Institute, University of Niigata, Niigata 951-8510, Japan; mac.oishi@mac.com

**Keywords:** Langerhans cell histiocytosis, pituitary stalk thickening, central diabetes insipidus

## Abstract

Central diabetes insipidus (CDI) is a rare disease in children and has a variety of etiologies. The major causes of CDI with pituitary stalk thickening (PST) are germinoma, Langerhans cell histiocytosis (LCH), and Lymphocytic infundibulo-neurohypophysitis, which are difficult to differentiate by imaging and require pathological diagnosis. We report a case of infantile-onset isolated neurohypophyseal LCH diagnosed by pathological findings. A 2-year-old girl presented with polydipsia and polyuria. CDI was diagnosed and treatment with oral desmopressin was initiated. Magnetic resonance imaging (MRI) of the head showed PST and absence of high-signal intensity of posterior pituitary on T1-weighted images. Follow-up MRI scans showed that the tumor mass was gradually increasing and extending posteriorly toward the area near the mamillary body. Simultaneously, anterior pituitary dysfunction was observed. She underwent a biopsy of the PST and LCH was diagnosed by immunohistochemical analysis. DNA analysis showed no BRAF V600E mutation. Monotherapy with 2-Chlorodeoxyadenosine reduced the tumor size but did not improve pituitary function. Isolated neurohypophyseal LCH should be considered in infantile-onset cases of CDI with PST. 2-CdA treatment resulted in rapid PST shrinkage. Further cases are needed to determine whether early diagnosis and treatment can prevent anterior pituitary dysfunction.

## 1. Introduction

Central diabetes insipidus (CDI) is characterized by the decreased release of antidiuretic hormone, resulting in a variable degree of polyuria. It occurs rarely in children and has a variety of etiologies, including genetic, traumatic, inflammatory, and neoplastic diseases [1]. Magnetic resonance imaging (MRI) shows pituitary stalk thickening (PST) in some patients with CDI. The main causative diseases of CDI with PST include germ cell tumors (GCTs), Langerhans cell histiocytosis (LCH), and lymphocytic infundibulo-neurohypophysitis (LINH), which are often difficult to differentiate [2,3,4,5,6]. According to a meta-analysis [7], the pooled proportion of PST caused by neoplasm in children was 67.4%, and the most common neoplasm was GCTs (26.9%), followed by LCH (22.2%).

In addition to brain MRI, tumor markers and other systemic examinations can help distinguish between GCTs, LINH, and LCH. GCTs can be detected using alpha-fetoprotein (AFP), β-human chorionic gonadotropin (β-hCG), and placental alkaline phosphatase (PLAP) in serum and cerebrospinal fluid (CSF) [8]; LINH by anti-rabphilin-3A antibodies [9]; and LCH by skeletal survey and skin examinations [2,10,11]. In case of the presence of lesions other than PST or any findings that raise the possibility of other diseases, biopsy of the PST can be avoided.

LCH is a rare histiocytic disorder characterized by the clonal proliferation of Langerhans cells. Its clinical manifestations and course are highly variable, ranging from occurrence in a single organ that is self-healing to multiple organ involvement [12]. CDI is a representative central nervous system (CNS) manifestation of LCH, and patients with multisystem and craniofacial involvement at diagnosis carry a significantly increased risk of developing CDI compared to those who do not [13]. However, the diagnosis of LCH is difficult in patients with a single lesion of the pituitary stalk. There are no known diagnostic biomarkers for LCH; therefore, diagnosis is confirmed by pathological examination of the lesion [14].

Reports of the isolated neurohypophyseal LCH are rare, especially in early childhood, and its clinical presentation is not clear. We report a case of isolated neurohypophyseal LCH, which developed into CDI and was diagnosed during the 1-year-follow-up period by biopsy of the enlarged PST.

## 2. Case Presentation

The patient was a Japanese girl aged 2 year and 9 months who presented with a sudden onset of polydipsia and polyuria. She was the first child of non-consanguineous parents and was born at full-term via normal delivery with a birth weight of 3186 g. Neonatal screening test results were within the normal range. She neither had previous health issues nor any abnormalities in her growth and development. There was no family history of polydipsia or polyuria.

The patient was conscious at presentation and exhibited no neurological abnormalities. Abnormal vital signs, physical findings of dehydration, and eczematous lesions on the skin were absent. Blood tests on admission showed normal renal function, electrolytes concentrations, and plasma osmolality. Urinalysis showed negative glycosuria and low osmolality of 141 mOsm/kg.

Her daily urine volume was up to 7000 mL/m^2^/day. She was diagnosed with CDI based on the water deprivation test and 1-deamino-8-D-arginine vasopressin (DDAVP) challenge test. The brain MRI scan showed PST and loss of physiological hypertense signal in the posterior pituitary gland on gadolinium-enhanced T1-weighted image (Figure 1a). There was no hyperintensity of basal ganglia and/or cerebellar nuclei on T2 FRAIR associated with LCH-related neurodegeneration. An arginine loading test confirmed normal secretion of growth hormone (Table 1), and other anterior pituitary functions, such as thyroid function test, prolactin level, and morning fasting cortisol levels, were within normal range. Treatment for CDI was initiated using nasal DDAVP 1.0 µg/day in two doses.

Whole-body bone radiographs suggestive of LCH were not observed. The results for tumor markers to distinguish intracranial germinoma, namely serum β-hCG, AFP, and PLAP in the CSF, were all within the standard ranges. The patient tested negative for anti-rabphilin-3A antibody, a potential marker for LINH. We performed MRI of the head every 3 months to check on the progression of PST over a period, during which gradual enlargement of the tumor mass and a posterior extension toward the area near the mamillary body was observed (Figure 1b–d). Moreover, anterior pituitary dysfunction were also evident (Table 1), with decreased levels of free thyroxine and insulin-like growth factor 1 (IGF-1) to 0.77 ng/dL (normal: 1.13–1.56 ng/mL) and 27 ng/mL (normal: 40–227 ng/mL), respectively. Her growth rate had declined to 4.4 cm per year. The actual cortisol levels were expected to be even lower in the presence of hypothyroidism. Therefore, we decided to supplement with a small dose of hydrocortisone once in the morning (3 mg/day) before levothyroxine sodium supplementation (25 µg/day). A biopsy of the PST using a ventricular endoscope was performed to confirm the diagnosis. LCH was diagnosed based on the positive results of the immunohistochemical analysis for S-100, CD68, and CD1a (Figure 2). Analysis of DNA extracted from biopsy tissue showed no V600E mutation in the BRAF by the Sanger sequencing.

The patient had a single-system, single site-type LCH. However, considering that the tumor exhibited a tendency to increase in size, and the anterior pituitary function was impaired, the patient was treated with 2-Chlorodeoxyadenosine (2-CdA) monotherapy at 5 mg/m^2^ for 5 consecutive days, every 4 weeks for six courses, which has been reported to be effective in treating LCH in CNS lesions [15]. The MRI findings after six courses of 2-CdA showed that the tumor had shrunk (Figure 1e). We will continue to follow up closely to confirm the appearance of LCH with neurodegeneration.

## 3. Discussion

We report a case of infantile-onset, isolated neurohypophyseal LCH, diagnosed by pathological findings of an enlarged lesion from the pituitary stalk over the course of the first year after CDI onset, based on longitudinal head MRI follow-up. The patient exhibited impaired anterior pituitary function over time. 

Differentiating between LCH, GCTs, and LINH causing CDI with PST is challenging. LCH can affect multiple organ systems, such as the skin and bones. If there is no lesion other than thickening of the pituitary stalk, diagnosis is extremely difficult. CSF and serum tumor markers, such as β-HCG, AFP, and PLAP, are useful for diagnosing GCT and avoiding biopsy [8]. A previous study showed that the sensitivity and specificity for detecting germinoma were 94% and 97%, respectively, with a cut-off value of 30 pg/mL for PLAP in CSF [16]. Furthermore, LINH should be distinguished as a cause of CDI with PST. Recently, a specific biomarker, namely rabphilin-3A, has proven to be a major auto-antigen in patients with LINH, and anti-rabphilin-3A antibodies are reported to exhibit high sensitivity and specificity [9]. Moreover, cases of childhood-onset rabphilin-3A antibody-positive LINH have been reported [17]. In our case, despite performing all these tests, none of the findings could lead to a diagnosis. Recently, positron emission tomography/computed tomography has been reported to have high sensitivity and specificity for staging and follow-up of pediatric patients with LCH, but further studies are needed to determine whether it is also useful in differentiating these diseases [18]. Considering the need for treatment, we performed a biopsy and diagnosed LCH by immunohistochemical analysis.

Reports of isolated neurohypophyseal LCH are limited. The cases of isolated suprasellar lesion LCH patients in children and adolescents are summarized in Table 2 [3,19,20,21,22,23,24]. The median age of onset was 10 years (range, 2–18 years), with no gender difference. All the patients developed CDI, and approximately 70% had hypopituitarism as well. The site of involvement was the pituitary stalk in all cases. There was extension into the hypothalamus in three cases and into the pituitary gland in one case. In all but one, the tumor shrunk with or without treatment, and prognosis was good.

The standard therapy for LCH of intracranial lesions, including pituitary stalk, or meningeal lesions (except local reaction to a skull vault lesion) is chemotherapy with vinblastine and steroid [25]. 2-CdA, which can cross the blood–brain barrier, is considered an indication for the hypothalamic–pituitary axis LCH. Dhall et al. [15] found that of 12 patients treated with 2-CdA for CNS mass lesions, eight had complete responses and four had sustained partial responses; 11 of the 12 patients remained in continuous remission or were progression free. Permanent sequelae of CNS LCH, such as panhypopituitarism, CDI and neurocognitive dysfunction, are not reversible. Our patient also exhibited rapid shrinkage of the mass after 2-CdA treatment, although CDI and hypopituitarism did not improve.

BRAF V600E is the most commonly identified mitogen-activated protein kinase (MAPK) activating mutation in LCH, and approximately half of cases carry this mutation [24]. The BRAF V600E mutation in cases of pediatric LCH is associated with high-risk features including relapse, high disease activity, primary refractory disease, and permanent sequelae [26,27]. In cases of isolated neurohypophyseal LCH, Nellan et al. [24] reported the frequency of BRAF V600E mutations to be 12.5% among eight children, while that reported by Huo et al. [3] was 50% among six children and adults. The frequency of the mutation is lower compared with that in cases of typical LCH and may be molecularly distinct from that in multisystem or other isolated organ LCH, the clinical significance of which requires further study.

## 4. Conclusions

Isolated neurohypophyseal LCH should be considered in infantile-onset cases of CDI with PST. 2-CdA treatment resulted in rapid PST shrinkage. Further cases are needed to determine whether early diagnosis and treatment can prevent anterior pituitary dysfunction.

## Figures and Tables

**Figure 1 children-09-00716-f001:**
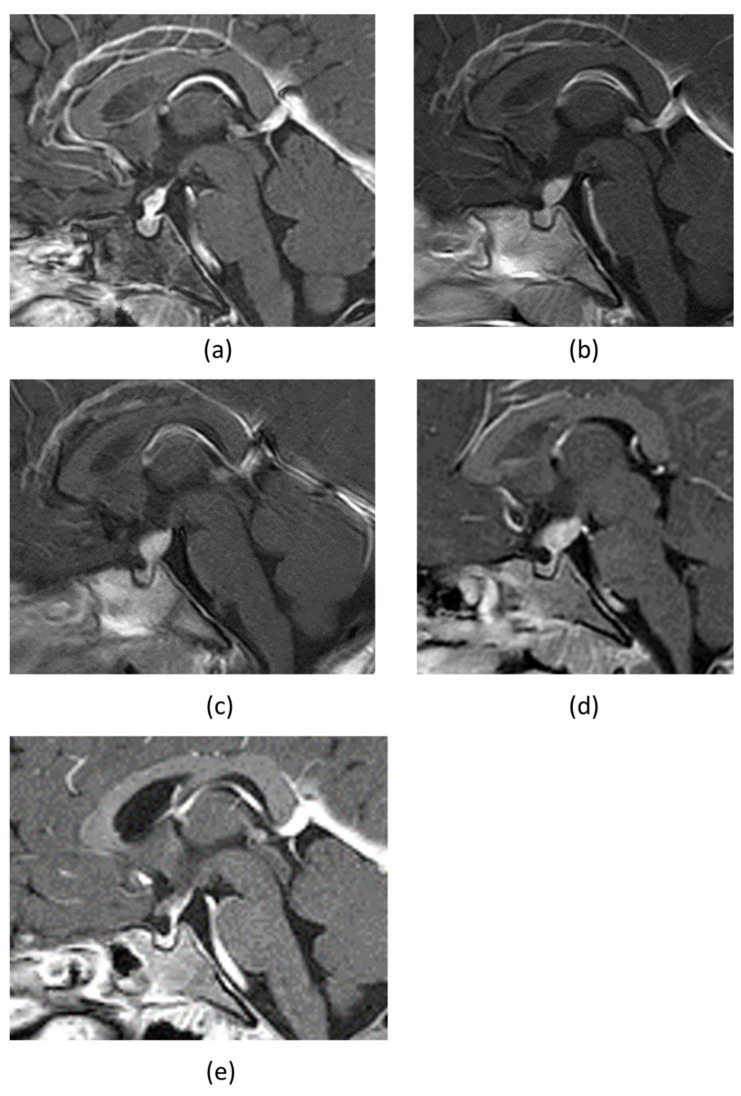
(**a**) The gadolinium-enhanced sagittal T1-weighted MRI of the brain at the time of admission, which showed pituitary stalk thickening and the loss of the physiological hypertense signal in the posterior pituitary gland. MRIs after three months (**b**), six months (**c**), and one year and three months (**d**), showed that the tumor mass was gradually increasing and seemingly moving posteriorly, reaching the area near the mamillary body. (**e**) MRIs after 6 courses of cladribine showed that the tumor tended to shrink.

**Figure 2 children-09-00716-f002:**
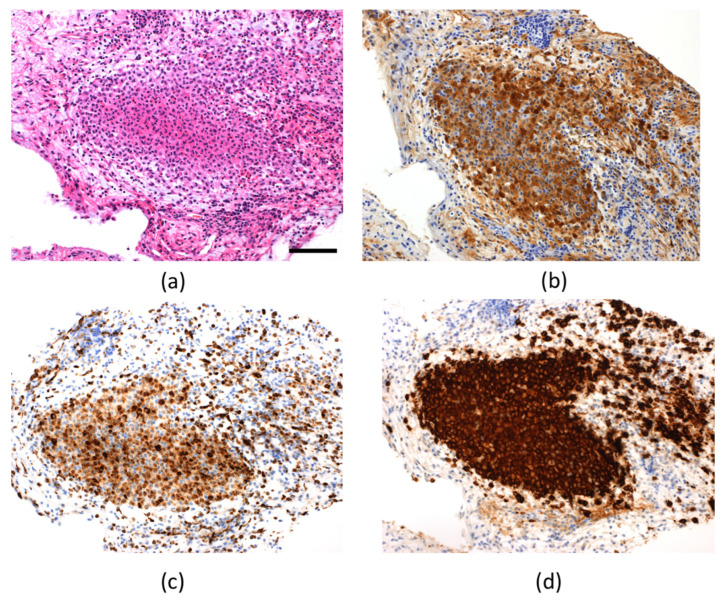
Histopathological examination of the pituitary stalk (**a**) Hematoxylin and eosin staining (scale bar; 100 µm); positive result of immunohistochemical analysis for S-100 (**b**), CD68 (**c**), and CD1a (**d**).

**Table 1 children-09-00716-t001:** The results of the anterior pituitary stimulation tests.

Loading Agents		Time	0	15	30	60	90	120	Assessment ^c^
	Hormones								
Arginine	GH ^a^ (ng/mL)		1.32	-	3.91	6.18	3.04	1.50	Normal
Insulin	Glucose ^b^ (mg/dL)		55	29	35	43	45	69	
Insulin	GH ^b^ (ng/mL)		0.2	0.2	0.2	0.9	0.8	1.7	Low
Insulin	ACTH ^b^ (pg/mL)		18.2	20.6	31.2	29.8	24.2	32.8	
Insulin	Cortisol ^b^ (µg/dL)		7.1	10.2	13.1	15.6	16	14.9	Low
TRH	TSH ^b^ (µIU/mL)		2.6	11.1	15.5	15.6	15.3	14.6	Delayed
TRH	PRL ^b^ (ng/mL)		28.1	57.6	72	57.1	63.3	45.2	Excessive
LHRH	LH ^b^ (mIU/mL)		<0.2	<0.2	0.2	<0.2	<0.2	<0.2	Low
LHRH	FSH ^b^ (mIU/mL)		1	3.1	5	5.4	5.3	4.9	Normal

GH, growth hormone; ACTH, adrenocorticotropic hormone; TRH, thyrotropin releasing hormone; TSH, thyroid-stimulating hormone; PRL, prolactin; LHRH, luteinizing hormone releasing hormone; LH, luteinizing hormone; FSH, follicle stimulating hormone. ^a^ Data of arginine loading test at the time of diagnosis of central diabetes insipidus (CDI). ^b^ Data of triple loading test (insulin, TRH and LHRH) at 15 months after CDI onset. ^c^ The data show decreased peak response of GH, LH and cortisol, delayed response of TSH, and excessive response of PRL.

**Table 2 children-09-00716-t002:** Summary of isolated suprasellar lesion LCH patients with CDI in children and adolescents after 2000.

Author (Year)	Age (Years)/Sex	DI before Treatment	APD before Treatment	Sites of Involvement	Treatment	Prognosis	References
Marchand, I. et al., (2011)	13/male	+	-	PS	Mass resection	Shrinkage	[19]
	16/female	+	+	PS	VBL, steroid/ Rx/2-CdA	Progression followed by regression	
	8/male	+	+	PS	VBL	Shrinkage	
	12/female	+	+	PS	VBL, steroid	Shrinkage	
Isoo, A. et al., (2000)	14/female	+	-	PS	Rx 20 Gy	Shrinkage	[20]
	9/female	+	*n*/A	PS, H	Rx 20 Gy	Shrinkage	
Nagasaki, K. et al., (2009)	13/female	+	-	PS	No Tx	Spontaneous shrinkage	[21]
Horn, E.M. et al., (2006)	18/male	+	+	PS	Rx 21 Gy	*n*/A	[22]
Kurtulmus, N. et al., (2015)	16/male	+	+	PS	Rx	Shrinkage	[23]
Huo, Z. et al., (2016)	9/female	+	+	PS	Rx	Shrinkage	[3]
	12/female	+	+	PS, H	No Tx	Spontaneous shrinkage	
	13/male	+	+	PS, H	Rx	Shrinkage	
	15/male	+	-	PS, *p*	No Tx	No change	
Zhou, W. et al., (2019)	17/male	+	-	PS	No Tx	No recurrence	[4]
Nellan, A. et al., (2020)	7/male	+	*n*/A	PS	VBL, prednisone	No recurrence	[24]
	4/female	+	*n*/A	PS	VBL, prednisone	No recurrence	
	9/male	+	*n*/A	PS	VBL, prednisone	Recurrence	
	10/male	+	*n*/A	PS	VBL, prednisone	No recurrence	
	3/male	+	*n*/A	PS	VBL, prednisone	No recurrence	
	2/female	+	*n*/A	PS	VBL, prednisone	No recurrence	
	7/female	+	*n*/A	PS	VBL, prednisone	No recurrence	
	8/female	+	*n*/A	PS	Rx	No recurrence	
Present case	2/female	+	+	PS	2-CdA	Shrinkage	

*n*/A, not available; DI, diabetes insipidus; APD, anterior pituitary dysfunction; LCH, Langerhans cell histiocytosis; PS, pituitary stalk; H, hypothalamus; *p*, pituitary; VBL, vinblastine; Rx, radiation therapy, 2-CdA, 2-Chlorodeoxyadenosine; Gy, gray.

## Data Availability

The data presented in this study are available upon request from the corresponding author. The data are not publicly available due to privacy restrictions.

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
