# Peer review of "Infantile-Onset Isolated Neurohypophyseal Langerhans Cell Histiocytosis with Central Diabetes Insipidus: A Case Report"

_children, 2022, doi:10.3390/children9050716_

Round 1

Reviewer 1 Report

The authors describe a meaningful case of pituitary stalk localized LCH leading to diabetes insipidus

the introduction is clear, the results are well presented and the images are of high quality.

I would like the authors to enrich the final conclusions with two points

first they might insist on the current added value of FDG PET-CT in the initial evaluation of LCH and cite the paper of Jessop et al in PBC 2020

second, I think it is of utmost importance to mention the need for long term follow-up of these patients who can develop years later CNS degenerative lesions, especially when they are BRAF mutated but not exclusively  (https://doi-org.proxy.insermbiblio.inist.fr/10.1002/pbc.26784)

Author Response

  • Responses to the comments of Reviewer 1

 We wish to express our appreciation to the reviewer for his or her insightful comments, which have helped us significantly improve the paper.

Comment 1: They might insist on the current added value of FDG PET-CT in the initial evaluation of LCH and cite the paper of Jessop et al in PBC 2020

Response: Thank you for your valuable your comments.

We have added the following description of PET/CT and additional references cited on page 6, Lines 140-143, in the discussion.

‘‘Recently, positron emission tomography/computed tomography has been reported to have high sensitivity and specificity for staging and follow-up of pediatric patients with LCH, but further studies are needed to determine whether it is also useful in differentiating these diseases (Jessop et al. in PBC 2020).’’

Comment 2: I think it is of utmost importance to mention the need for long term follow-up of these patients who can develop years later CNS degenerative lesions, especially when they are BRAF mutated but not exclusively  (https://doi-org.proxy.insermbiblio.inist.fr/10.1002/pbc.26784)

Response: We have added a comment that we will continue to follow up closely to confirm the appearance of LCH with neurodegeneration in Page 5, Lines 122-123. 

Thank you again for your comments on our paper. We trust that the revised manuscript is suitable for publication.

Reviewer 2 Report

It is a well-written case report. Nevertheless, I require some clarifications from the authors:

  • Line 92 Why such a small dose of Hydrocortisone? 4.6mg/m2? Is it 3 mg each dose, or daily? Replacement therapy should be around 10 mg/m2 even in central adrenal insufficiency (at least 7.5mg/m2/die). Please provide reference for such a small dose, if confirmed.
  • It might be useful to know if – in patient’s brain MRI - T2/FLAIR hyperintensity of basal ganglia and /or cerebellar nuclei were found, and how they changed after chemotherapy. These have been described in association with neurodegenerative LCH (McClain KL et al, CNS Langerhans Cell Histiocytosis: Common Hematopoietic Origin for LCH-Associated Neurodegeneration and Mass Lesions, cancer 2018, and ref #5)

Additionally:

Please pay attention to grammar mistakes such as:

 line 18 "requires" should become "require")

Line 87 "gradually" should become "gradual"; "extended" should be  "extension"

Line 88 “was” should become “were”

Author Response

  • Responses to the comments of Reviewer 2

We wish to express our appreciation to the reviewer for his or her insightful comments, which have helped us significantly improve the paper.

Comment 1: Line 92 Why such a small dose of Hydrocortisone? 4.6mg/m2? Is it 3 mg each dose, or daily? Replacement therapy should be around 10 mg/m2 even in central adrenal insufficiency (at least 7.5mg/m2/die). Please provide reference for such a small dose, if confirmed.

Response: If the HPA axis is completely impaired, as you suggest, then hydrocortisone 10mg/m2/day is required. However, the patient had only a slight impairment of the HPA axis with a basal cortisol level of 7.1 μg/dL and a peak level of 16.0 μg/dL on an insulin load test. However, it has been reported that in the presence of hypothyroidism, serum cortisol does not decrease even in the presence of ACTH deficiency(Endocr J. 1999;46(1):183-186). In the present case, the cortisol level was actually expected to be even lower because the patient had central hypothyroidism with FT4 0.77 ng/dL and TSH 2.6 μIU/mL. Therefore, we decided to supplement with a small amount of hydrocortisone once in the morning before levothyroxine sodium replacement.We added an explanation to Lines 93-95 regarding the interpretation of cortisol values.

Comment 2: It might be useful to know if – in patient’s brain MRI - T2/FLAIR hyperintensity of basal ganglia and /or cerebellar nuclei were found, and how they changed after chemotherapy. These have been described in association with neurodegenerative LCH (McClain KL et al, CNS Langerhans Cell Histiocytosis: Common Hematopoietic Origin for LCH-Associated Neurodegeneration and Mass Lesions, cancer 2018, and ref #5)

Response: Thank you for your significant suggestion. I did check the patient's T2/FRAIR for hyperintensity of the basal ganglia and /or cerebellar nuclei, but did not see the finding. There was no such finding after chemotherapy either.

I have added following comment in the case presentation on Page 2, Lines 78-79.

‘‘There was no hyperintensity of basal ganglia and/or cerebellar nuclei on T2 FRAIR associated with LCH-related neurodegeneration.’’

Comment 3: Please pay attention to grammar mistakes such as:

Line 18 "requires" should become "require")

Line 87 "gradually" should become "gradual"; "extended" should be "extension"

Line 88 “was” should become “were”

Response: We have corrected the grammatical errors you have pointed out. We again obtained an English review by a native English speaker and have attached a certificated proof.

Thank you again for your comments on our paper. We trust that the revised manuscript is suitable for publication.

Reviewer 3 Report

I would like to thank the authors for this interesting and well-presented case report. Some suggestions for the manuscript follow:

  1. Line 50: instead of “range”, ranging
  2. Line 52: instead of “nerves”, nervous
  3. Line 54: instead of “than”, compared to
  4. Line 58: since the child in the case report is 2 years 9 months old, instead of “infancy”, early childhood should be written
  5. Line 64: “a non-consanguineous parents”: either “a consanguineous couple” or “non-consanguineous parents”
  6. Line 87: “gradual enlargement of the tumor mass and a posterior extension toward the area near the mamillary body, was observed”
  7. Line 90: growth rate instead of “height growth rate”
  8. Line 90: in order to identify a secondary hypothyroidism, both fT4 and TSH values should be given (without TSH values, a primary hypothyroidism with a possible reactive TSH increase could be the case)
  9. Line 92: in order for the initiation of hydrocortisone treatment to be justified in the text, levels of cortisol and ACTH or the synacthen test results should be first described
  10. Line 99: shows instead of show
  11. Line 100: …in the posterior pituitary gland
  12. Line 108: you should better describe what tests were used for the given hormone results. For example, cortisol and prolactin levels were the results of which tests? And what do you mean by insulin test? In addition, I would suggest you better correlate specific test results with the corresponding tests in the table given so that it is easier for the reader to identify the tests performed and to evaluate the results
  13. Line 114: …considering that the…
  14. Line 121: We report
  15. Line 122: …over the course of the…
  16. Line 126: multiple organ systems
  17. Line 132: rabphilin-3A antigen and not antibody
  18. Line 135: In our case
  19. Line 159: mutations to be 12.5%

Author Response

  • Responses to the comments of Reviewer 3

We wish to express our appreciation to the reviewer for his or her insightful comments, which have helped us significantly improve the paper.

Comments to the Author:

I would like to thank the authors for this interesting and well-presented case report. Some suggestions for the manuscript follow:

  1. Line 50: instead of “range”, ranging
  2. Line 52: instead of “nerves”, nervous
  3. Line 54: instead of “than”, compared to
  4. Line 58: since the child in the case report is 2 years 9 months old, instead of “infancy”, early childhood should be written
  5. Line 64: “a non-consanguineous parents”: either “a consanguineous couple” or “non-consanguineous parents”
  6. Line 87: “gradual enlargement of the tumor mass and a posterior extension toward the area near the mamillary body, was observed”
  7. Line 90: growth rate instead of “height growth rate”
  8. Line 90: in order to identify a secondary hypothyroidism, both fT4 and TSH values should be given (without TSH values, a primary hypothyroidism with a possible reactive TSH increase could be the case)
  9. Line 92: in order for the initiation of hydrocortisone treatment to be justified in the text, levels of cortisol and ACTH or the synacthen test results should be first described
  10. Line 99: shows instead of show
  11. Line 100: …in the posterior pituitary gland
  12. Line 108: you should better describe what tests were used for the given hormone results. For example, cortisol and prolactin levels were the results of which tests? And what do you mean by insulin test? In addition, I would suggest you better correlate specific test results with the corresponding tests in the table given so that it is easier for the reader to identify the tests performed and to evaluate the results
  13. Line 114: …considering that the…
  14. Line 121: We report
  15. Line 122: …over the course of the…
  16. Line 126: multiple organ systems
  17. Line 132: rabphilin-3A antigen and not antibody
  18. Line 135: In our case
  19. Line 159: mutations to be 12.5%

Response: We have corrected the phrase, the grammatical errors, or typos you have pointed out. (Comments 1, 2, 3, 4, 5, 6, 7, 10, 11, 13, 14, 15, 16, 17, 18, 19).

We again obtained an English review by a native English speaker and have attached a certificated proof.

Response to Comment 8, 9: The FT4 levels are concurrent with the TRH loading test, and the TSH levels are listed in Table 1. The data suggest central hypothyroidism. The patient had only a slight impairment of the HPA axis with a basal cortisol level of 7.1 μg/dL and a peak level of 16.0 μg/dL on an insulin load test. However, it has been reported that in the presence of hypothyroidism, serum cortisol does not decrease even in the presence of ACTH deficiency(Endocr J. 1999;46(1):183-186). In our case, the cortisol level was actually expected to be even lower because the patient had central hypothyroidism with FT4 0.77 ng/dL and TSH 2.6 μIU/mL. Therefore, we decided to supplement with a small amount of hydrocortisone once in the morning before levothyroxine sodium replacement. We added an explanation to Lines 93-95 regarding the interpretation of cortisol values.

Response to Comment 12: Thank you for your comments.

We have added the corresponding loading agent name and assessment to Table 1.

Thank you again for your comments on our paper. We trust that the revised manuscript is suitable for publication.